# Is Pupil Response to Speech and Music in Toddlers with Cochlear Implants Asymmetric?

**DOI:** 10.3390/audiolres15040108

**Published:** 2025-08-14

**Authors:** Amanda Saksida, Marta Fantoni, Sara Ghiselli, Eva Orzan

**Affiliations:** 1Institute for Maternal and Child Health—IRCCS “Burlo Garofolo”–Trieste, via dell’Istria 65/1, 34137 Trieste, Italy; 2Educational Research Institute, Gerbičeva 62, 1000 Ljubljana, Slovenia; 3AUSL Piacenza—Guglielmo da Saliceto Hospital, Via Antonio Anguissola 15, 29121 Piacenza, Italy

**Keywords:** ear advantage, cochlear implants, toddlers, pupillometry, listening effort, speech perception, music perception, auditory lateralization

## Abstract

**Background:** Ear advantage (EA) reflects hemispheric asymmetries in auditory processing. While a right-ear advantage (REA) for speech and a left-ear advantage (LEA) for music are well documented in typically developing individuals, it is unclear how these patterns manifest in young children with cochlear implants (CIs). This study investigated whether pupillometry could reveal asymmetric listening efforts in toddlers with bilateral CIs when listening to speech and music under monaural stimulation. **Methods:** Thirteen toddlers (mean age = 36.2 months) with early bilateral CIs participated. Pupillary responses were recorded during passive listening to speech and music stimuli, presented in quiet or with background noise. Each child was tested twice, once with only the left CI active and once with only the right CI active. Linear mixed-effects models assessed the influence of session (left/right CI), signal type (speech/music), and background noise. **Results:** A significant interaction between session and signal type was observed (*p* = 0.047). Speech elicited larger pupil sizes when processed through the left CI, while music showed no significant lateralized effects. Age and speech therapy frequency moderated pupil responses in speech and music trials, respectively. **Conclusions:** Pupillometry reveals subtle asymmetric listening effort in young CI users depending on the listening ear, suggesting early emerging functional lateralization despite sensory deprivation and device-mediated hearing.

## 1. Introduction

### 1.1. REA and LEA

Ear advantage (EA) is an auditory processing phenomenon where individuals demonstrate superior performance for a competing stimulus when presented to one ear compared to the other. A small right EA (REA) is typically observed for speech sounds, indicating the left hemisphere dominance for processing complex auditory stimuli characterized by rapid transitions and complex acoustic features, such as speech [1,2,3]. Conversely, left EA (LEA) has been shown for musical, environmental, and animal sounds, implying left-hemisphere dominance for processing music, which may involve more steady-state or tonal qualities [1,4]. The adult-like patterns of ear advantages have also been found in dichotic tasks with normally hearing newborns and young infants, and even in newborn hearing screening results with the click-based Transient Evoked Otoacoustic Emissions (TEOAE) test [3,4,5,6], indicating an innate/anatomical/perceptual right-ear advantage for complex stimuli. However, recent studies indicate that REA may also be influenced by cognitive factors such as memory, attentional focus, and cognitive load/listening effort and may serve as a coping mechanism in complex auditory environments, such as dynamic cocktail party situations [7,8,9].

The prevalence and magnitude of REA for speech stimuli appears to increase through early childhood until age 5 or 6, after which it remains relatively stable [3,5].

### 1.2. REA and LEA in the Context of Hearing Loss

In the context of hearing loss (HL), findings appear to be similar to those observed in normally hearing populations. For example, in cases of single-sided deafness, a right-ear advantage (REA) was found for speech audiometry results but not for pure-tone stimuli [10].

Importantly, a recent study showed a pronounced REA for speech in children with early simultaneous bilateral cochlear implants (CIs). Similarly to findings in a normal-hearing population, the study demonstrated that REA was influenced by age and auditory experience in preschool children with CIs. These results suggest early and simultaneous implantation may facilitate typical auditory pathway development [11,12].

Nonetheless, REA appears particularly difficult to measure with dichotic listening tasks in children with significant hearing loss, possibly due to immature attention, limited working memory, or other developmental factors impacting vocabulary and language acquisition. This difficulty has led to observations of limited evidence for EA in patients with HL, although these findings cannot be entirely disentangled from underlying developmental factors [13,14]. Potentially because of this, dichotic perception in children with significant hearing loss is poor.

Similarly, the potential left-ear advantage (LEA) for music may be difficult to assess in patients with significant HL, particularly in those with CIs. Hearing loss and hearing devices affect music perception, appreciation, and participation, and the extent and nature of these effects vary greatly between individuals [15,16]. While dichotic listening conditions have been shown to improve music perception in CI users significantly [17], it remains unclear how such improvements relate to ear advantages in processing different types of sounds and how ear advantages could be exploited as coping mechanisms in complex auditory environments.

### 1.3. Pupillometry as an Index of Listening Effort

Complex auditory environments increase listening effort, and pupil dilation has emerged as a sensitive indicator of subtle variations in this phenomenon, recently becoming a benchmark method for its assessment [18,19]. Given the challenges encountered when measuring ear advantage (EA) with subjective dichotic listening tasks in children with hearing loss (HL), described above, pupillometry may offer a more precise way to study EA and the influence of cognitive factors, such as memory and attentional focus, on EA.

To date, only one study has observed EA through pupillometry. The results showed that when young adults listened dichotically, attending to the right ear typically required less effort (as indicated by smaller pupil size), even when accuracy scores were comparable. Moreover, the right-ear advantage for reduced listening effort became more pronounced during more difficult tasks [20].

### 1.4. Research Questions

The aim of this study was to determine whether pupillometry provides a viable method for establishing the presence of an ear advantage (EA) in young children with cochlear implants (CIs). Specifically, we investigated whether pupil size changes could be observed without requiring an active task when children listened monaurally to speech or music under both optimized and adverse listening conditions. Demonstrating such changes would provide further evidence that EA can be modulated by attention and listening effort even in young CI users.

To this end, a group of toddlers with congenital hearing loss (HL) and early bilateral CIs was tested in two sessions: once with the right CI switched off and once with the left CI switched off. We hypothesized that listening to speech with only the left CI activated would elicit greater listening effort, reflected by increased pupil size. Conversely, we expected that listening to music with only the right CI activated would similarly increase listening effort and pupil size. Additionally, we anticipated that background noise would interfere with these processes, given the documented difficulties young children face in attending to speech under adverse acoustic conditions. Based on prior findings in the same children (with both CIs activated), where pupil size differences during speech trials were observed only in the absence of background noise [21], we hypothesized that a potential right-ear advantage (REA) for speech would be evident only in trials without background noise. Finally, we analyzed ear differences in relation to their hearing, cognitive, and language skills and in relation to the ear that was implanted first. We hypothesized that language skills and implantation side for the first CI may influence EA effects.

## 2. Methods

### 2.1. Participants

The study included a cohort of children (Table 1) with CIs (*N* = 13 children, 7 girls, *mean age* = 36.2 months, *range*: 16.73–47.03) with congenital hearing loss and early sequential bilateral CIs. Included in the cohort were preschool children who had successfully received implants before 24 months old, who were at least 1 month post their second CI fitting, who were regularly followed up at the clinic, and who showed psychophysical abilities and language development within the normal range. For toddlers in the present cohort, the first CI was on the right side for 12 out of 13 children and was implanted within the first two years of life. They were regularly followed at the clinic; their mean-aided threshold was always fitted to ≤35 dB HL. Language comprehension scores (CDI) were 5° to 90° at the time of testing; all children had normal cognitive abilities and motor skills (BSID-III) and no visual deficits. None of the children were bi-modally (oral and sign language) bilingual. The final size of the group was limited by the number of children who met the above criteria and who we had access to at the institute where the study was conducted. Participants came from various regions in Italy and had diverse socio-economic backgrounds. The same cohort, but with two additional toddlers, was tested in a previous study with the same stimuli and the same procedure, but with both implants switched on [21]. All parents were informed about the purpose of the study and the testing methods and gave their written consent for their children to participate before the testing began. Participants were treated in accordance with the Ethical Principles for Medical Research Involving Human Subjects (WMA Declaration of Helsinki). The regional medical Ethical Committee approved the study on 25 September 2018, ID: 2490; the testing was conducted in 2019 and 2020. The participant sample is described in more detail in Appendix A.

### 2.2. Stimuli

Target and noise auditory stimuli were the same as in the previously published study [21]: (1) 5 s long recordings of spoken rhymed verses recorded by a native Italian female actress using child-directed speech; (2) 5 s long excerpts from the instrumental version of “The Happy Song”; and (3) multi-channel and multi-speaker speech noise, with signal-to-noise ratios (SNRs) as follows: no noise = only signal played; low noise = 10 dB SNR; high noise = 0 dB SNR [22,23,24]⁠. For the purpose of the study, the signal-to-noise levels were re-coded into quiet and noise (including both low and high noise ratios).

The visual stimulus was again the same as in the previously published study [21]: a single 5 s long excerpt from a silenced version of an animated film Koyaa. The excerpt was unrelated to the speech input and was repeated in all trials. A total of 27 still frames from the same film were presented during the inter-stimulus intervals.

### 2.3. Procedure and Apparatus

The experimental procedure and apparatus were again repeated from the previously published study [21]. Each child was tested in a quiet experimental room in two sessions, once with the left implant switched on and once with the right implant switched on (sessions L and R). The order of the sessions was counterbalanced among participants. The testing sessions were separated by at least 1 month to ensure toddlers were attentive to stimuli during both sessions. The total duration of each testing session was ca. 5 min. During the test, children passively listened to the trials containing music or speech (in pseudo-randomized orders), with or without background noise, while watching the visual stimulus on the eye-tracking screen. Figure 1 presents the schematic presentation of the trial structure.

Toddlers’ gaze and pupil size were recorded with a TOBII T120 eye-tracker (Tobii AB, Stockholm, Sweden) at 60 Hz, integrated into a 17-inch TFT screen. Infants were seated on their parents’ lap at about 60 cm from the screen. Parents kept their eyes shut during the test, but their ears were not blocked. The experiment was presented through PsyscopeX B88 software.

The 3D audio simulation created and implemented for the study is presented in Figure 1B and described in more detail in the previously published study [21] (Figure 1B).

### 2.4. Preprocessing and Statistical Analysis

The prepossessing included the removal of non-physiological artifacts (data-points larger than 8 and smaller than 2 mm), the selection of valid data-points (measured when gaze was fixated on the screen), and the exclusion of trials in which more than 30% of the pupil data were missing during the test phase [25,26]. This led to the exclusion of 28% of the recorded trials. Furthermore, we removed data from participants who only had valid pupil measurements in one session. Thus, 9 out of initial 13 participants were included in the final data sample. The missing data in the included trials were linearly interpolated. Pupil data were baseline-corrected using subtractive baseline correction based on the average of the first 300 ms at the beginning of each test phase. All analyses and visualizations were programmed in R 4.4.1 [27,28]⁠⁠. The anonymized dataset, including raw pupil data and external variables, is publicly available, along with the analysis script, at https://osf.io/f7s68/?view_only=cf08ebfbc66343c5b0a53329c1cd6a9c (accessed on 1 August 2025). Pupil size was then averaged per trial (1000–5000 ms) based on the permutation tests (based on the restricted likelihood ratio test statistic) for mixed models [29].

The possible presence of EA for speech and music, as reflected by the difference in the average pupil size, was evaluated with a linear mixed-effects model using the lmer function from the lme4 package in R 4.4.1. [30]. The model M1 included session (left ear, right ear), signal (music, speech), background (quiet, noise), and their interactions as fixed factors, with a random intercept for subject to account for repeated measures. Possible interactions between signal type and session were evaluated with a separate model.

The possible effects of age, hearing age, aided hearing threshold, language proficiency, and cognitive factors on pupil behavior during the two listening sessions were assessed with separate linear mixed models that examined the possible interaction between these factors and the session. Model details are available in Appendix A.

## 3. Results

Prior to analysis, the distributions of pupil size values per session, signal, and background were visually inspected via histograms and Q-Q plots. No major deviations from normality were observed. Descriptive analysis of the values based on nine participants with remaining valid pupil data in both sessions is presented in Table 2.

The model M1 showed a significant interaction effect of signal and session on the pupil size, as measured by the analysis of deviance of the model (Type II Wald chi-squared tests) (*χ^2^*(1) = 3.93, *p* = 0.047). Model predictions and the underlying data are presented in Figure 2B. The addition of the fixed factors and their interactions to the model M1 did not significantly improve model fit compared to the null model (*χ*^2^(7) = 6.81, *p* = 0.45), making the evidence that the fixed factors explained variability in pupil size weak. Nonetheless, visual inspection of residual plots showed the predicted heteroscedasticity and low autocorrelation (independence), indicating that the model is valid in predicting the effects (Figure 2C). As is also visible in Figure 2A, the presence of background noise did not significantly change pupil response. In fact, independently, the three fixed factors did not significantly explain the variability of the pupil size outcome in M1. However, the significant interaction between session and signal factors justified the creation of two subset models, for speech and music trials. While the model for music trials showed no significant effect, the model for speech trials (M1a) showed a significant effect of session on pupil size (*β* = 0.08, *SE* = 0.94, *t* = 1.99, *p* = 0.049). Residual diagnostics indicated no major violations of model assumptions. Model comparison using a likelihood ratio test indicated that the model with session as a fixed effect provided a significantly better fit than the null model (*χ^2^*(1) = 3.97, *p* = 0.046), indicating that the fixed factor indeed explains the variability in pupil size.

In the next step, the interaction effects of session and age, hearing age, aided hearing threshold, language proficiency, speech therapy sessions/week, and cognitive factors on pupil behavior were assessed for speech trials (M2) and music trials (M3).

The model M2 showed only a significant effect of age on pupil size during speech trials (*β* = 0.02, *SE* = 0.01, *t* = 2.09, *p* = 0.039; *R^2^adj* = 0.19, *F*(15,86) = 2.59, *p* = 0.002). Residual diagnostics indicated no major violations of model assumptions.

The model M3 showed a significant interaction effect of session and the number of speech therapy sessions/week during music trials (*β* = 0.44, *SE* = 0.18, *t* = 2.38, *p* = 0.019; *R^2^adj* = 0.19, *F*(15,84) = 2.55, *p* = 0.004; Analysis of variance: *χ^2^*(1) = 3.04, *p* = 0.019). Residual diagnostics indicated no major violations of model assumption. Model predictions for M2 and M3 overlaid on the actual data, and the respective residual plots are presented in Figure 3A,B.

While, due to the lack of variance, the side of first implantation could not be included in the above models, the visual inspection of individual EA during speech trials (Figure 3C) reveals that the EA, as reflected in pupil size, is not consistent across participants and that pupil behavior in the participant (03) with the left ear implanted first resembles that of the majority of the participants.

## 4. Discussion

The present study investigated asymmetric pupil responses to speech and music in toddlers with cochlear implants (CIs) as a potential index of ear advantage (EA). Using pupillometry as a non-invasive measure of listening effort, we observed that pupil size varied as a function of the listening ear and the type of auditory stimulus. Specifically, a significant interaction between stimulus type (speech vs. music) and the session (left CI active vs. right CI active) was found, suggesting the presence of asymmetric auditory processing patterns in young children with CIs.

### 4.1. Asymmetric Auditory Processing in Toddlers with Cochlear Implants

The observed asymmetry in pupil responses supports the hypothesis that even young children with early sequential bilateral cochlear implantation exhibit lateralized auditory processing for speech and music. In line with prior findings in normal-hearing populations [1,3], speech stimuli elicited greater listening effort (indexed by larger pupil size) when the left CI (typically associated with right-ear input in hearing individuals) was active. Conversely, increased pupil dilation was observed for musical stimuli when listening with the right CI, although to a lesser degree. These findings extend previous research suggesting that REA for speech is an early-emerging and robust phenomenon [4,6,31] and indicate that typical patterns of hemispheric specialization can be observed even in populations with early congenital hearing loss, with bilateral auditory input restored within the first three years of life.

The results align with studies showing that early and simultaneous or closely sequential implantation supports typical auditory pathway development [31]. Importantly, the order of closely sequential implantation did not seem to play a crucial role in this cohort: the response of the one child who had the left ear implanted first is aligned with the majority of children’s responses. Thus, in this single case, the order of implantation did not crucially define the ear or cortical advantages for speech processing. More general conclusions are nonetheless not possible based on a single case. It is noteworthy, however, that EA is not observable in all children’s pupil response but represents a trend in the majority of the participants. Such inconsistency in pupil dilation has been observed elsewhere and may be the consequence of possible individual differences in the attentional capture or uncontrolled factors that may have influenced the pupil dilation [32].

The absence of a strong LEA for music may again be explained by the specific characteristics of the tested cohort. Music perception in CI users is known to be highly variable and generally less robust than speech perception [15,16], also due to the technological limitations of CIs in transmitting complex musical features such as pitch and timbre. The weaker lateralization effect for music could reflect both device-related factors and the developmental stage of the participants.

### 4.2. Listening Effort and Cognitive Load

Increased pupil dilation when listening to speech stimuli with left CI active in our cohort may indeed reflect increased listening effort during listening with the non-advantageous ear [18,19]. This is consistent with prior observations in adults where attention to the non-dominant ear increased listening effort [20]. However, contrary to our expectations, background noise did not significantly modulate pupil size. Pupil size difference between left and right sessions even in the presence of background noise contrasts with previous results [21] and may indicate that toddlers with CIs allocate more effortful attention to speech when listening with the disadvantageous ear but are only able to exploit favorable signal-to-noise ratios effectively in non-noisy listening conditions, possibly due to immature auditory scene analysis abilities. However, more research on speech perception in noise in infants and toddlers with CIs is needed for a more conclusive interpretation of this result.

Moreover, the interaction between age and pupil response indicates that maturational factors play a significant role in speech perception with CIs. Older toddlers exhibited larger pupil responses during speech trials in both testing sessions, suggesting either increased cognitive engagement or greater awareness of the listening task demands, specifically when listening to speech; for music trials in this study, the effect was not visible. The absence of the effect for music trials could of course be attributed to the small sample size. It could, however, be also attributed to the nature of the auditory input. For comparison, in a recent study that included normally hearing infants aged 8 to 48 months who listened to warble sounds, no effect of age on pupil size was found [32].

Interestingly, during music trials, the number of weekly speech therapy sessions interacted with session effects on pupil size. This finding may imply that intensive auditory–verbal therapy enhances speech processing and could also influence broader auditory attention mechanisms, including the perception of non-speech sounds.

### 4.3. Methodological Considerations

This study demonstrates the feasibility of using pupillometry as a sensitive, rapid, and non-invasive tool to investigate auditory processing and cognitive effort in very young children with hearing loss. Compared to traditional dichotic listening tasks, which require active participation and cognitive maturity, passive pupillometry measures offer an accessible alternative for this population.

Nevertheless, study limitations may limit broader implications. Above all, the relatively small sample size, although consistent with previous infant pupillometry studies, limits the generalizability of the findings. The variability in participants’ age, hearing history, and language abilities may have introduced additional noise into the data. Additionally, the design involved monaural stimulation by deactivating one implant, which does not perfectly mimic naturalistic listening environments where binaural integration occurs, or dichotic listening environments that elicit enhanced monoaural attention.

### 4.4. Clinical and Theoretical Implications

Clinically, these results suggest that ear-specific listening effort profiles might be useful in tailoring auditory rehabilitation strategies in young CI users. An understanding of individual asymmetry patterns could inform personalized interventions, aiming to support auditory development more effectively. Theoretically, the findings contribute to a growing body of evidence indicating that hemispheric specialization for auditory processing is preserved despite early auditory deprivation, if the restoration of hearing with cochlear implantation occurs early in life. They also emphasize the role of cognitive effort as an integral component of auditory processing, even from early childhood.

Future research should aim to replicate these findings in larger cohorts, including comparisons with age-matched normal-hearing controls, and to explore the developmental trajectory of ear advantages and listening effort across the preschool years. Investigating the relationship between neural markers of auditory processing (e.g., via EEG or fNIRS) and pupillometric measures would provide deeper insights into the neural mechanisms underlying these behavioral effects.

## 5. Conclusions

This study shows that toddlers with early bilateral cochlear implants exhibit asymmetric listening effort for speech and music, detectable through pupillometry. A right-ear advantage for speech processing suggests early development of hemispheric specialization despite congenital hearing loss. Pupillometry offers a valuable, non-invasive tool for studying auditory processing in young CI users, with implications for personalized rehabilitation strategies. Further research is needed to confirm these findings in larger cohorts and across developmental stages.

## Figures and Tables

**Figure 1 audiolres-15-00108-f001:**
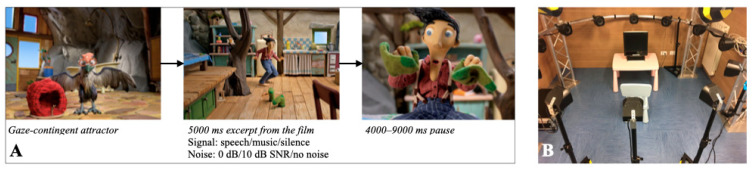
(**A**) The schematic representation of the trial structure, with the task preceded by an attractor and followed by a pause. (**B**) The experimental setup.

**Figure 2 audiolres-15-00108-f002:**
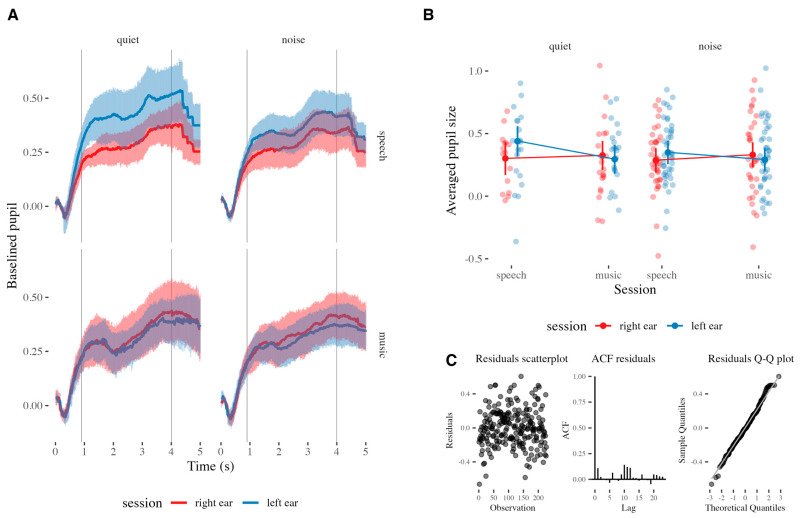
(**A**) The average time-course of pupil responses to speech and music against quiet and noisy backgrounds when listening monoaurally. (**B**) Term predictions of the model (dots connected by solid lines) with the underlying actual data (jittered dots). Error bars represent confidence intervals (95%). (**C**) The scatterplot, autocorrelation plot, and Q-Q plot of the residuals of the model.

**Figure 3 audiolres-15-00108-f003:**
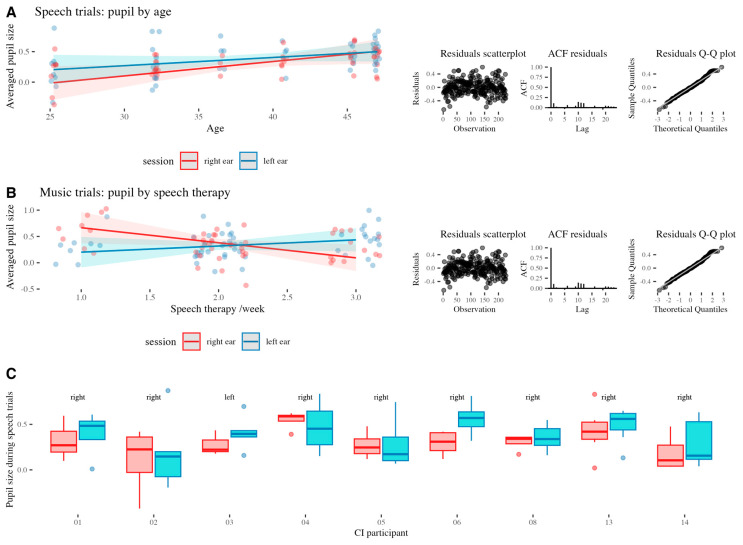
(**A**) The model M2 predictions, underlying actual data, and residuals plots for speech trials. (**B**) The model M3 predictions, underlying actual data, and residuals plots for music trials. (**C**) EA in individuals with CI during speech trials in relation to the ear implanted first (text above graphs).

**Table 1 audiolres-15-00108-t001:** Participants’ personal and anamnestic data.

Participant ID	Sex	Bimodally Bilingual (1 = Yes)	Aided Threshold	First CI Age	Second CI Age	Side of First CI	IQ Verbal	IQ Nonverbal	IQ Motor	Age (Months)	Time from First CI	Speech Therapy Sessions/Week	Language Perception Score (Percentile)
01	m	0	25	11.93	15.77	right	89	105	97	40.77	28.83	2	50
02	f	0	31	13.77	19.63	right	83	110	97	25.23	11.47	3	50
03	f	1	31	20.67	25.17	left	71	90	100	36.53	15.87	2	90
04	f	0	27	29.5	40.3	right	86	110	97	47.03	17.53	2	50
05	f	1	31	10.7	12.77	right	97	110	107	32.2	21.5	2	90
06	m	0	34	21.73	29.37	right	65	96	94	46.93	25.2	3	5
07	f	0	34	26.03	29.13	right	59	85	85	30	3.97	2	5
08	m	1	32	18.97	33.67	right				39.7	20.73	3	5
09	m	0	28	12.03	12.17	right	69	100	82	18.57	6.53	2	5
11	f	0	27	11.83	14.8	right	86	95	91	16.73	4.9	3	10
12	f	1	31	11.07	15.5	right	86	97		31.87	20.8	2	10
13	m	0	26	20.9	28.9	right	95	122		45.33	24.43	1	50
14	m	0	31	10.73	24.47	right	94	105	97	32	21.27	2	50

**Table 2 audiolres-15-00108-t002:** Basic descriptive analysis of the pupil size values per session, signal, and background.

Signal	Session	Background	Trials (N)	Pupil Size Mean	Pupil Size SD	Pupil Size Median	Pupil Size Min	Pupil Size Max
Speech	Right ear	Quiet	15	0.30	0.13	0.35	0.03	0.48
Speech	Right ear	Noise	37	0.30	0.25	0.29	−0.39	0.84
Speech	Left ear	Quiet	19	0.44	0.27	0.52	−0.19	0.85
Speech	Left ear	Noise	41	0.37	0.25	0.33	−0.10	0.85
Music	Right ear	Quiet	21	0.34	0.27	0.29	0.00	0.99
Music	Right ear	Noise	33	0.34	0.28	0.33	−0.30	0.82
Music	Left ear	Quiet	21	0.33	0.21	0.33	−0.02	0.71
Music	Left ear	Noise	40	0.32	0.25	0.31	−0.04	0.91

## Data Availability

The dataset is presented for publication for the first time, and we are not aware of other comparable or duplicate published study. The anonymized dataset including raw pupil data is publicly available, along with the analysis script, at https://osf.io/f7s68/?view_only=cf08ebfbc66343c5b0a53329c1cd6a9c (accessed on 1 August 2025).

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
