# Peer review of "Is Pupil Response to Speech and Music in Toddlers with Cochlear Implants Asymmetric?"

_audiolres, 2025, doi:10.3390/audiolres15040108_

Round 1
Reviewer 1 Report
Comments and Suggestions for Authors
This is an interesting and clear manuscript that uses pupillometry to study how toddlers with cochlear implants process speech and music differently in each ear. Using passive listening with pupil measurements provides a useful, non-invasive way to look at how young children pay attention and how much effort they use when listening. The study is well-explained and adds essential knowledge about how the brain processes sounds in early childhood.
The research examines whether listening effort, measured by pupil size, shows a preference for one ear (ear advantage) in toddlers with cochlear implants. 13 toddlers listened to speech and music sounds through one ear at a time. The results show a right-ear advantage for speech, seen by dilated pupil responses when the left implant was active, which suggests early brain specialization. The authors have appropriately addressed key limitations, including the small sample size, which restricts the generalizability of the findings. Variability in participants’ age, hearing history, and language abilities may also contribute to data variability. Additionally, monaural stimulation by deactivating one implant simultaneously does not fully replicate natural binaural listening conditions.
However, a few clarifications and minor improvements are needed before the manuscript is suitable for publication:
Background noise had no significant impact on pupil size, supporting the notion of asymmetric auditory processing patterns in young children with cochlear implants. To improve readability, this point could be more clearly explained.
On line 40, please provide the full form of the abbreviation “TEOAE” for clarity.
Author Response
We thank the reviewer for the positive review of the manuscript and hope it will be of interest of the broad readership in the field. Below, we answer to the two comments
Comment 1: Background noise had no significant impact on pupil size, supporting the notion of asymmetric auditory processing patterns in young children with cochlear implants. To improve readability, this point could be more clearly explained.
Response 1: Thank you for this comment. It is true that the response is somewhat difficult to interpret in the light of the previously published study, and in the light of the fact that we do not have extensive evidence of how do infants and toddlers with CIs manage background noise and to what extent does it affect their attention and, above all, learning mechanisms. We have tried to be more clear both about the hypotheses and about the interpretation of the results in the revised manuscript.
The following text was added / modified:
[Based on prior findings in the same children (with both CIs activated), where pupil size differences were observed only in the absence of background noise, we hypothesised that a potential right-ear advantage (REA) for speech would be evident only in trials without background noise.] (Lines 97-99)
[Increased pupil dilation when listening to speech stimuli with left CI active in our cohort may indeed reflect incresed listening effort during listening with the inadvantageaus ear [19], [20]. This is consistent with prior observations in adults where attention to the non-dominant ear increased listening effort [22]. However, contrary to our expectations, background noise did not significantly modulate pupil size. Pupil size difference between left and right sessions even in the presence of background noise contrasts previous results [23] and may indicate that toddlers with Cis allocate more effortful attention to speech when listening with the inadvantageaus ear, but are only able to exploit favorable signal-to-noise ratios effectively in non-noisy listening conditions, possibly due to immature auditory scene analysis abilities. However, more research on speech perception in noise in infants and toddlers with Cis are needed for a more conclusive interpretation of this result.] (lines 249-257
Comment 2: On line 40, please provide the full form of the abbreviation “TEOAE” for clarity.
Response 2: Thank you for this note. we clarified it in the text now.
Reviewer 2 Report
Comments and Suggestions for Authors
really a great paper!
Author Response
Dear Reviewer, many thanks, we hope that it will receive attention of the readership in the field and that we will be able to continue research on ear advantages in CI children in the future.
Reviewer 3 Report
Comments and Suggestions for Authors
The paper has shown elegantly the persistence of REA in the first years with CI I staking place and should guide CI side decision for implanting.
The selection criteria is well covered and has taken into account potential outliers to provide a more statistical soundness.
Just one question. Have you found any child answering contrary? What could that indicate? IS a child with a LEA?
Author Response
Dear Reviewer,
Thank you for your positive comments. We are looking forward to new studies that will shed even more light into the pupillary correlates to neural processes in children with HI.
Comment 1: Just one question. Have you found any child answering contrary? What could that indicate? IS a child with a LEA?
Response 1: This is an excellent question. In fact, we have initially planned the additional analysis of the one child with the left CI implanted first, but have then concentrated on the group effects instead. In any case, we have now added the individual analyses, which clearly show that the REA is not entirely consistent across participants and that the outlier child behaved just like majority of the children who received the right CI first. We have added individual plots that clearly show it to the Figure 3, and the following text:
[Finally, we analysed ear differences in relation to their hearing, cognitive, and language skills, and in relation to the ear that was implanted first. We hypothesised that language skills and implantation side for the first CI may influence the EA effects.] (lines 100-102)
[While, due to the lack of variance, the side of first implantation could not be included in the above models, the visual inspection of individual EA during speech trials (Figure 3C) reveals that the EA, as reflected in pupil size, is not consistent across participants, and that pupil behavior in the participant (03) with the left ear implanted first resemples that of the majority of the participants. ](lines 222-225)
[Figure 3C EA in individuals with CI during speech trials in relation to the ear implanted first (text above graphs).] (line 228)
The discussion points were amended as follows:
[Importanty, the order of closely sequential implantation did not seem to play a crucial role in this cohort: the response of the one child who had the left ear implanted first is aligned with the majority of children’s responses. Thus, in this single case, the order of implantation did not crucially define the ear or cortical advantages for speech processing. More general conclusions are nonetheless not possible based on a single case. It is noteworthy, however, that EA is not observable in all children’s pupil response, but represents a trend in the majority of the participants. Such inconsistency in pupil dilation has been observed elsewhere and may be the consequence of possible individual differences in the attentional capture, or uncontrolled factors that may have influenced the pupil dilation [34].] (lines 248-255)